# Changes in Body Mass, Physical Activity, and Dietary Intake during the COVID-19 Pandemic Lockdowns in Canadian University Students

**DOI:** 10.3390/biology12020326

**Published:** 2023-02-17

**Authors:** Madison Bell, Markus J. Duncan, Karen A. Patte, Brian D. Roy, David S. Ditor, Panagiota Klentrou

**Affiliations:** 1Department of Kinesiology, Brock University, St. Catharines, ON L2S 3A1, Canada; 2Children’s Hospital of Eastern Ontario, Ottawa, ON K1H 8L1, Canada; 3Department of Health Sciences, Brock University, St. Catharines, ON L2S 3A1, Canada

**Keywords:** COVID-19, university students, body mass index, physical activity, caloric intake, macronutrient intake, micronutrient intake

## Abstract

**Simple Summary:**

This paper looks at changes in weight, physical activity, and food/dietary intake during the first year of the COVID-19 pandemic in undergraduate students enrolled in a Canadian university. The investigation used an online survey format to collect data at the end of the two lockdown periods over one year. Participants were recruited from universities across Canada. The results show a modest weight gain accompanied by a decrease in physical activity and an unhealthy change in diet. Specifically, students reported that their engagement in specific physical activities and overall activity levels decreased during the lockdowns. The number of calories also decreased over time, while the foods consumed decreased in nutritional value. Thus, the modest weight gain observed may have resulted from the decrease in physical activity and diet quality. In conclusion, Canadian food and physical activity recommendations were not met during the lockdowns caused by the pandemic, which highlights the physical and nutritional implications of confinement.

**Abstract:**

This study examined changes in body mass and body mass index (BMI), physical activity, and dietary intake in Canadian university students during the first year of the COVID-19 pandemic. Two self-reported recall surveys were conducted: after the first lockdown in September 2020 (T1) and following the second lockdown in March 2021 (T2). Eligible participants were full-time undergraduate students attending a Canadian university and residing in Canada during the first year of the pandemic. At T1, 510 students (99 male, 411 female) completed the survey, and of those, 135 (32 males, 103 females) completed the survey at T2 (73% attrition). At both T1 and T2, most participants were 18–24 years of age (93% and 90%, respectively), Caucasian (73% and 78%, respectively), and resided in the province of Ontario (79% and 80%, respectively). Body mass increased from T1 to T2 (+0.91 ± 3.89 kg *t*(132) = −2.7, *p* = 0.008). BMI also increased from T1 to T2 (+0.30 ± 1.33 kg/m^2^ [*t*(130) = −2.5, *p* = 0.012), with a greater number of participants within the overweight range (19.8% versus 24.4%, respectively). At T1, 38% of the participants reported a decrease in physical activity, while the number of students reporting a decrease in activity increased to 56% at T2. Dietary energy intake decreased from 1678 ± 958 kcal/day at T1 to 1565 ± 842 kcal/day at T2 [c^2^(1) = 7.2, *p* = 0.007]. Diet quality also decreased, with participants not meeting the recommended daily allowance for essential macro and micronutrients. A decrease was observed in daily servings of fruits (−27%, *p* < 0.001), vegetables (−72%, *p* < 0.001), and grains (−68%, *p* < 0.001). In conclusion, despite a small decrease in dietary energy intake, a modest weight gain occurred during the first year of the COVID-19 pandemic in this cohort of Canadian university students, which was potentially related to decreased physical activity and diet quality.

## 1. Introduction

On 11 March 2020, the World Health Organization (WHO) declared the rapidly spreading coronavirus disease 2019 (COVID-19) a global pandemic. The first COVID-19 case in Canada was discovered on 20 January 2020 [1,2]. The virus spread through most provinces in Canada by the end of March 2020 as the first wave of lockdowns was enforced, with a second wave of the virus and enforced lockdowns occurring in the winter of 2021. As the number of daily cases reached record levels [3], various preventative measures were imposed across Canada, including physical distancing protocols, quarantines, curfews, mask and vaccine mandates, and closures of nonessential businesses. Consequently, Canadian universities closed their facilities and shifted to an online teaching format; undergraduate students were forced into home confinement for weeks, potentially increasing the risk of adopting sedentary behaviors [4,5].

Students pursuing university undergraduate education are already a population susceptible to weight gain [6,7,8], which may pose chronic long-term health risks, as once excess weight is gained, it is difficult to lose [9]. In previous studies, first-year university students were six times more likely to gain weight than age-matched community controls [10], and females enrolled in first-year university were 3.8 times more likely to gain weight than their same-age counterparts not attending university [11]. These changes have been attributed to the transition to living away from home for the first time and adopting unhealthy lifestyle habits. Healthy dietary habits are shown to decline in students adjusting to independent living [7,12]; for instance, increases in energy intake and alcohol consumption, erratic eating hours, and decreased quality of nutrient-dense foods have been found [13]. A reduction in the quality and quantity of physical activity and regular exercise has also been reported, especially in first-year university students [14]. For example, before the COVID-19 pandemic, 81.2% of Canadian students in Quebec (*N* = 1989) were found not to meet the Canadian Physical Activity Guidelines for Adults (Canadian Society for Exercise Physiology, 2011) of 150 min of moderate to vigorous physical activity per week [15].

Similar patterns during the early COVID-19 pandemic lockdowns have been found in postsecondary students around the world, with studies generally reporting an increase in snacking and a decrease in both physical activity and diet quality [5,16,17,18,19,20]. Indeed, early evidence supports the negative impact that COVID-19 has on physical activity. In one multicontinental study, the frequency, duration, and intensity of physical activity decreased by 35%, 34%, and 43%, respectively, during home confinement [16]. A similar finding was also observed in a Canadian study, whereby 90% of the university student participants who were meeting the guidelines before the pandemic became less active during the pandemic [19]. A study of Italian university students observed a significant increase of total time spent in sedentary behaviors and sleeping/napping during the first COVID-19 lockdown and a significant decrease of total physical activity, while body mass index (BMI) did not change [5]. Evidence from an Italian sample also indicated an increase in inadequate nutrition during the first lockdown, with males significantly increasing the consumption of takeaway food and alcohol [21]. A meta-analysis found that both body mass and BMI significantly increased in the general population (i.e., adolescents and adults >16 years of age) during the first lockdown of the pandemic [22]. Additionally, one study of university students comparing before to during the first COVID-19 lockdown in Saudi Arabia found that energy intake increased significantly (2876.4 ± 158.2 to 2684.3 ± 172.6 kcal/day), and 32% of participants reported having gained weight since before the pandemic; however, fast food and fried food intake decreased, 22% lost weight, and 46% maintained their weight [20]. Further research during the prolonged pandemic response and subsequent lockdowns is needed. There is limited longitudinal data on Canadian university students collected during both the 2020 and 2021 COVID-19 lockdown periods.

The aim of this study was to investigate potential changes in body mass and BMI, physical activity perceived change, and dietary intake during the first year-two lockdowns of the COVID-19 pandemic in Canadian university students. Based on previous combined evidence regarding lifestyle changes in university students prior to the pandemic as well as changes that have been reported during the pandemic, our hypothesis was that body mass and caloric intake would increase while physical activity and diet quality would decrease in university students during the first year of the pandemic-induced lockdowns.

## 2. Materials and Methods

### 2.1. Participants

Participants were recruited online during the summer of 2020 through social media postings and a general recruitment email sent to university administrators and department heads for distribution. Recruitment involved inviting students to participate in an online anonymous retrospective survey looking to examine changes in physical activity and dietary intake over the first 12 months of the pandemic. A total of 1346 students showed interest in the study by clicking the survey link distributed through social media and email announcements. This resulted in 49% of those, i.e., 510 students (n = 99 males, n = 411 females), completing the first online survey (T1) at the beginning of the fall academic term (August–September 2020). Of those, 135 (n = 32 male, n = 103 female) also completed the second survey (T2) near the end of the winter term (March 2021), resulting in an attrition rate of 73.5%. All participants confirmed their eligibility, which was to have attended a Canadian university for undergraduate studies on a full-time basis during the 2019/2020 academic year and be a minimum of 18 years of age. 

All participants were informed online about the details of the study and provided consent to participate. The study and all related procedures received ethical clearance initially from the Brock University Research Ethics Board and subsequently from the Research Ethics Boards of 23 other participating Universities: Acadia University, Carleton University, Dalhousie University, Lakehead University, Mount Royal University, Nipissing University, Simon Fraser University, Trent University, University of Alberta, University of British Columbia, University of Guelph, University of Fraser Valley, University of Lethbridge, University of Manitoba, University of New Brunswick, University of Ottawa, University of Regina, University of Saskatchewan, University of Waterloo, University of Windsor, University of Winnipeg, and York University.

### 2.2. Procedures and Assessments

The study involved a two-part survey administered twice (T1 and T2) using Qualtrics (XM, Provo, UT, USA). At T1 (i.e., beginning of the fall 2020 term), part 1 of the survey asked participants a range of demographic questions and to report current body mass and height. Part 2 of the survey asked participants to reflect on their perceived changes in physical activity as well as their dietary intake over the 6-month period from March to September 2020. At T2 (i.e., March 2021), part 1 of the survey asked participants the same demographic questions and to report anew their current body mass and height. Part 2 of the survey asked participants to reflect on their physical activity and dietary intake over the previous 6-month period from October 2020 to March 2021 to assess changes from T1 to T2 (Figure 1). Parts 1 and 2 of the survey can be viewed in Appendix A.

#### 2.2.1. Demographics 

At each data collection time, i.e., T1 and T2 (Figure 1), participants answered a range of demographic questions including sex, race/ethnicity, current age (years), year of study (year 1–5+, or recent graduate), where in Canada they lived (region/province), whom they lived with (friends, family, or alone), and what type of residence they lived in (house, apartment, or other). These questions were adapted from a previous study by Beaudry et al. [7]. Such questions have been previously used to examine the impact of living at home or away from home on diet and dietary habits [23] as well as physical activity [24]. Likewise, questions on social dynamics, such as living with parents or alone, have been previously used to examine their influence on dietary choices and intake [25]. The demographic characteristics of the participating cohort are presented in Table 1.

#### 2.2.2. Anthropometrics and Physical Activity

At each time point, participants also reported their current, self-assessed body mass (kg) and height (cm). BMI (kg/m^2^) was then calculated using the reported body mass and height. Classifications of BMI are made by weight class using the BMI range and comprise underweight (<18.5 kg/m^2^), normal (18.5 kg/m^2^–24.9 kg/m^2^), overweight (25 kg/m^2^–29.9 kg/m^2^), and obese (>30 kg/m^2^) [26].

Perceived change in physical activity was self-assessed. At T1, participants were asked to indicate whether relative to before the pandemic and their participation in each of the listed activities, including at-home workouts, walking, jogging, running, cycling, swimming, weight training, recreational sports, and sport-specific training, decreased (−1), increased (1), or did not change (0) from March to September 2020 (i.e., during the first lockdown). At T2, participants were asked to indicate whether relative to the previous 6 months (i.e., since T1), their participation in each of the same activities decreased (−1), increased (1), or did not change (0) from October 2020 to March 2021 (i.e., during the second lockdown). Overall physical activity change scores were then calculated based on the responses in all listed activities. These questions, although not validated, were designed to provide us with an idea of the participants’ perceived change in their physical activity directly related to the pandemic lockdowns. Compared to using standardized physical activity questionnaires, asking for change in activities does not require participants to retroactively recall exactly how much activity they were participating in before lockdowns but merely to report whether they had increased or decreased certain activities during the lockdowns. Similar questions asking for self-reported change in activity have been used to for retrospective surveys during COVID-19-related lockdowns [27,28,29,30].

#### 2.2.3. Food Frequency Questionnaire (FFQ)

The Block 2014 FFQ (NutritionQuest, Berkeley, CA, USA) was used to assess energy intake and diet quality for both T1 and T2. This FFQ included 127 food and beverage items and additional questions to determine fat, protein, carbohydrate, sugar, and whole grain consumption. The participants were given a portion size sheet to help them quantify the amounts of food that they were eating and to better aid them in answering the questions. Dietary habits were examined over the previous 6 months with a focus on estimating the consumption of macronutrients (carbohydrates, fats, proteins), micronutrients (vitamins and minerals), and overall mean energy intake (kcal/day). The FFQ is a commonly used measure of dietary intake and quality [31] and has been tested for reliability and validity in Canadian women [32]. Further comparisons were then made to Canadian recommendations [33] by using the mean of the specific nutrient (i.e., protein, carbohydrates, calcium, etc.) and dividing it by the recommended dietary allowance value multiplied by 100 to obtain a value percentage. The FFQ provided estimated daily servings for grains, meats, meat alternatives, fruits, and vegetables, which were also compared against Canada’s Food Guide regarding servings per day [33].

### 2.3. Statistical Analysis

#### 2.3.1. General Analysis and Data Assessment

Analyses were performed using R (R Core Team, 2020). Statistical significance was set at *p* = 0.05. Generalized linear regression models were used to evaluate change over time while controlling for confounding variables. Ordinal dependent variables were assessed using proportional odds logistic regression implemented in the ordinal package. Histogram inspection of energy intake and dietary intake of macro and micronutrients approximated a gamma distribution. In order to fit a generalized linear model with a gamma distribution and log link function, values of 0 were assumed to represent values below the detection threshold and replaced with 0.01. Multivariate outliers were assessed when modeling using regression diagnostic plots (Ordinal Q-Q plot and Cook’s distance plot for influence detection).

#### 2.3.2. Longitudinal Analysis

All analyses comparing T1 to T2 changes over time were performed using only the data of the 135 participants who completed both surveys. Changes in body mass and BMI from T1 to T2 were assessed using paired sample t-tests, with effect size determined by Cohen’s d for significant pairwise comparisons and interpreted based on the Cohen criteria: 0.2 = small, 0.5 = medium, 0.8 = large effects [34]. After fitting models, Type II sum of squares analysis of variance (ANOVA) was used to detect omnibus differences between sexes after controlling for other variables in the model. Generalized linear models with a gamma distribution function and a log link were used to assess the difference in energy intake as an outcome. A random intercept by the subject was used to account for the repeated measures nature of the data. Changes in macro and micronutrient intake and adequacy from T1 to T2 were assessed using paired sample t-tests.

Finally, regression analysis was used to assess the relationship between body mass, physical activity, and dietary intake changes. We applied a conditional change approach [35,36] where change scores (T2-T1) were regressed on baseline (T1) scores in addition to confounding variables (age, sex, ethnicity, province) before regressing on independent group variables. 

## 3. Results

There was a significant increase in the self-assessed body mass and BMI reported at the end of T1 versus the end of T2 (Table 2). Specifically, body mass increased on average by 0.91 ± 3.89 kg, [*t*(132) = −2.7, *p* = 0.008, 95%CI:(0.24, 1.58)], while the overall increase in BMI was 0.30 ± 1.33 kg/m^2^ [*t*(130) = −2.5, *p* = 0.012, 95%CI:(0.07, 0.53)]. In addition, although it appears that body mass significantly increased from T1 to T2 only in male students, after controlling for baseline body mass at T1, age category, ethnicity, and provinces/region, the Type II sum of squares ANOVA found no significant difference between males and females in terms of changes in body mass from T1 to T2 [F(1) = 0.08, *p* = 0.78]. Importantly, there was an increase from T1 to T2 in the number of participants meeting the criteria for overweight (19.8% to 24.4%, respectively) and a decrease in the number of students meeting the criteria for normal weight (61.7% to 54.8%, respectively) [26] (Figure 2).

Figure 3 shows the distribution of all individual responses about perceived changes in overall physical activity as calculated from the changes in specific activities. Figure 4 shows the percent frequency of responses regarding the overall physical activity and each of the specific activities reported at T1 and T2. Overall, a greater number of students reported decreasing their physical activity during the second lockdown (T2 survey) than during the first lockdown (Figure 3). Specifically, 48% of students reported increasing and 38% reported decreasing their physical activity during the first lockdown, while 56% of students reporting decreasing their physical activity and 31% reported increasing their physical activity during the second lockdown. The same pattern of a larger decrease reported at T2 for the period of the second lockdown was also found for each of the listed activities (Figure 4). 

After removing outliers, we found a small but significant decrease (−7%) in dietary energy intake over time [c^2^(1) = 7.2, *p* = 0.007]. Specifically, at T1, students had a mean energy intake of 1678 ± 958.54 kcal/day [95% CI: (1534, 1835)], while at T2, those students had a mean energy intake of 1565 ± 842.1 kcal [95% CI: (1430, 1712)] (Figure 5). Furthermore, the intake of most macro and micronutrients decreased from T1 to T2, with significant differences found in the daily intake (g/day) of carbohydrates (*p* < 0.001, d = 0.3), sugars (*p* ≤ 0.001, d = 0.4), fiber (*p* = 0.013, d = 0.2), and alcohol (*p* < 0.001, d = 0.1) (Figure 6A–D). In addition, daily servings of fruit decreased from 1.7 ± 1.4 at T1 to 1.2 ± 1.0 at T2 (−27%, *p* < 0.001, d = 0.2). The same was observed for daily servings of vegetables, which decreased from 2.0 ± 1.3 at T1 to 0.6 ± 0.9 at T2 (−72%, *p* < 0.01, d = 1.2). Likewise, daily servings of grains decreased from 5.7 ± 3.9 at T1 to 1.9 ± 1.7 at T2 (−68%, *p* < 0.001, d = 1.0). 

In terms of adequacy, protein intake was over the recommended daily allowance (RDA) for both T1 and T2. A similar overconsumption was also observed for carbohydrate intake, including sugar, at T1, with intakes remaining over the RDA, irrespective of the significant decrease at T2 [33]. In contrast, fiber, vitamin D, and calcium intake were lower than what Canadian guidelines describe as adequate [33] (Figure 7). 

Finally, the regression analysis showed that after controlling for body mass at T1, age, sex, ethnicity, and province/region, no significant association was found between body mass change (T2-T1) and demographic variables, including year of study, physical living arrangements (i.e., type of housing), and social living arrangements (i.e., with whom the respondent was living). Likewise, no significant correlation was found between body mass and changes in physical activity and dietary energy intake. In terms of perceived change in overall and specific physical activities, after controlling for all confounding variables, at T1, only cycling showed a significant difference even after Bonferroni correction for the nine types of activity assessed (*p* = 0.0007, *p_adjusted_* = 0.006). Specifically, students living in a house were more likely to increase cycling than those living in an apartment (AOR = 2.52, 95% CI: [1.47, 4.37]). Students living in a house were also more likely to increase swimming (AOR = 1.92, 95% CI: [1.16, 3.20]) and walking (AOR = 1.78, 95% CI: [1.08, 2.95]) compared to those living in an apartment, but these differences were not significant after correcting for multiple comparisons (*p* = 0.012, *p_adjusted_* = 0.10 for swimming; *p* = 0.024, *p_adjusted_* = 0.22 for cycling). In terms of social living arrangements, at T1, cycling also showed a significant difference before Bonferroni correction but not after (*p* = 0.013, *p_adjusted_* = 0.11). Specifically, students living alone were less likely to increase their cycling compared to both those living with family (AOR = 2.43, 95% CI: [1.13, 5.25]) and those living with friends/roommates (AOR = 1.27, 95% CI: [0.50, 3.24]). At T2, there was an insufficient number of individuals living alone, so separation models were repeated with individuals living alone excluded. Consequently, activity changes were not significantly related to social living arrangements. Finally, after controlling for the confounders and adjusting for multiple comparisons, changes in energy intake did not differ based on either physical activity changes or living arrangements, whether physical or social. 

## 4. Discussion

The present study aimed to investigate the changes in body mass, physical activity, dietary intake, and diet quality that occurred among undergraduate students in Canadian universities during the first and second lockdowns of the COVID-19 pandemic. The novel aspects of this study include the nationwide demographic and, most importantly, longitudinal examination of changes occurring across the first and second lockdown periods, building on previous studies that have primarily been cross-sectional in design and during the first lockdown. Of note, our specific cohort of university students included young adults in various years of study, mainly from Ontario, which resembles the general university population in Canada as most undergraduate students are young adults and the majority of Canadian universities are in Ontario. However, the study cohort was not representative in terms of ethnicity and sex as it consisted heavily of white female students (~80%). In addition, most of our participants were living with family in a house, which was also the typical arrangement during the pandemic, as across Canada, the pandemic restrictions led to the closures of schools and nonessential businesses and shifts to online learning both in the spring of 2020 and winter of 2021. Our hypotheses were partially confirmed. Canadian university students had a significant increase in self-assessed body mass and BMI reported in the first year of the pandemic (March 2020–March 2021), with more students classified in the overweight BMI category at T2 compared to T1. In addition, 48% of the students reported engaging in less physical activity during the first lockdown compared to before the pandemic, and this percentage increased to 56% during the second lockdown. However, contrary to our hypothesis, dietary energy intake (kcals/day) decreased from T1 to T2. There was also a decrease in most macronutrient and micronutrient intake as well as in diet quality when considering the amount and dietary adequacy of calcium, fruits, vegetables, and grains.

Despite a self-reported decrease in dietary energy intake over the T1 and T2 time points, students had a significant increase in self-assessed body mass and BMI (large effect size) reported at T1 versus T2, which is in line with the only other study examining similar parameters, in university students in Italy, during the first COVID-19 lockdown [5]. There was also an increase between T1 and T2 in the number of participants meeting the criteria for overweight (19.8% to 24.4%, respectively) compared to normal weight (61.7% to 54.8%, respectively). However, we have no data on these students from before the pandemic to examine any changes from pre-pandemic to the first lockdown, i.e., T1. Therefore, it is not clear if the changes in body mass, BMI, and diet during the COVID-19 response differ from what would be expected outside of the pandemic. As mentioned, previous studies indicate weight gain and dietary changes are common in this population and at this stage of life [6,7,8]. It is also recognized that engagement in regular physical activity helps with weight maintenance. Therefore, due to confinement, it was expected that students would not maintain their physical activity level, which was confirmed by the findings. Almost half of the students reported lower physical activity during the first lockdown, and over half of those who completed the survey at T2 reported decreasing their physical activity during the second lockdown. This observation is in line with other studies on student populations [5,19]. It suggests that the modest weight gain the students reported may be partly attributable to a decrease in physical activity, which resulted from the university closures and thus restricted access to recreation and athletic facilities and was also brought about by the implementation of remote learning and reduced mobility. Students were forced to live in home confinement, increasing the risk of reduced physical activity and of adopting sedentary behaviors [5]. 

Furthermore, when looking at the interaction of living arrangements with the reported change in physical activity, our findings show that students living in a house were more likely to increase their engagement in physical activities such as cycling, swimming, and walking than individuals living in an apartment, while individuals living with their family were also more likely to increase their cycling than individuals living alone. It is plausible that students living in apartments were more likely to lack the space or access to facilities to be active or to live in neighborhoods less conducive to physical activity than their peers living in a house. Alternatively, these observations may reflect social influences. Previous research has demonstrated that social support is a strong motivator to engage in physical activity, with positive social pressure from friends being the strongest motivator among males and social support from their family being the strongest motivator among females [37]. These results suggest that under similar circumstances, in order to reduce the risk of adopting sedentary behaviors, university-aged students should be encouraged to move in with others, should not be alone, and should be provided with physical proximity to outdoor recreational areas.

As in Bertrand et al. [19], our study found a significant decrease in total daily energy intake in these students. Additionally, our participants were below the recommended daily intake for both adult females (1750 to 2350 kcal/day) and adult males (2200 to 3000 kcal/day) [38]. Previous studies that assessed dietary intake using either 24 h recalls or food logs/diaries found no differences in dietary intake over the course of 1 year of university [39,40], whereas studies that have used FFQs found significant decreases in total energy intake (kcals) and macronutrients [14,41]. The FFQ used in the present study provided nutrient intake, the frequency of foods eaten, and a composite measure of the frequency and quantity of foods eaten represented in grams per day. In addition, by looking at grams of nutrients per day, we could determine diet quality, which is an umbrella term used to describe how well an individual’s diet conforms to dietary recommendations while remaining healthy, balanced, and nutritious [42]. Compared to the RDA, our participants reported higher protein and carbohydrate intakes, especially sugar, at both T1 and T2, despite a decrease between time points. At the same time, they reported lower than the RDA intakes of important micronutrients, including calcium and vitamin D, which are essential for bone health, blood clotting, and nerve transmission [43,44] especially for individuals in their early 20s as this is when peak bone mass is achieved. Fiber was also lower than what Canadian guidelines describe as adequate intake and decreased further from T1 to T2. Previous studies [45] have found a link between fiber intake, weight management, and obesity. Fiber-rich foods tend to be more satiating due to their lower energy density as compared with low-fiber foods such as digestible polysaccharides and simple sugars [46]. Relatedly, our study found significant decreases in the consumption of fruits, vegetables, and grains. Specifically, we found our participants consuming on average 0.5 servings of fruits and vegetables and 1.7 servings of grains, which is considerably lower than the 7–10 servings of fruits and vegetables and 6–8 servings of grains a day recommended by the 2007 Canada Food Guide [33]. This decrease in consumption of fruits and vegetables may help to explain the increase in body mass observed since it is well documented that fruit and vegetable intake, as well as fiber intake, have inverse relationships with body mass [47,48]. Our results are consistent with previous research in North American university students during the COVID-19 pandemic, which also reported a decrease in overall diet quality, along with decreased consumption of fruits, vegetables, and nuts [19]. Interestingly, alcohol intake also decreased in our study, in both male and female students, contradicting previous studies looking at the effect of the pandemic that found alcohol consumption had increased during lockdowns in adults [19,21]. On the other hand, our finding is in line with the COMPASS study, which reports a decrease in alcohol consumption during the COVID pandemic in adolescents, suggesting that younger people tend to be more social with their drinking [49].

This study has several limitations. The main limitation is the small size of our sample, which consisted heavily of white females and thus was not entirely representative of the student population in Canadian universities. Thus, the results of this study cannot be generalized and should be interpreted with caution. Likewise, the T1 response rate and high attrition rate between T1 and T2 may introduce selection bias, which also makes the results not generalizable to all Canadian university students as students that participated may differ in weight and diet from their peers that did not respond or dropped out after T1. In addition, since the study was conducted under pandemic-induced restrictions, body mass, height, changes in physical activity, and diet were self-assessed and self-reported by the participants and thus are subject to recall error and social desirability bias. Another intrinsic limitation of this study is the lack of baseline pre-pandemic data, which could not be collected as the pandemic was not foreseen. Lastly, we were unable to control for all relevant confounders; in particular, energy expenditure was not measured. Instead, perceived changes in physical activity were reported in a binary fashion.

## 5. Conclusions

In conclusion, a modest weight gain occurred between the first and second lockdowns of the COVID-19 pandemic in this cohort of Canadian university students. This weight gain occurred despite a modest decrease in total energy intake and was potentially related to the decline in physical activity and diet quality. Future research is needed to assess dietary changes during the ongoing pandemic and into the recovery period to determine if changes are sustained or were temporary in response to the lockdown. This study adds to the evidence that many emerging adults fall short of nutritional recommendations, particularly for fiber, calcium, and vitamin D, increasing their risk for future chronic disease.

## Figures and Tables

**Figure 1 biology-12-00326-f001:**
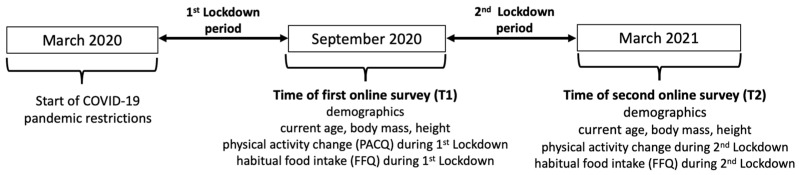
Timeline of data collection. Both first (T1) and second (T2) data collection times included demographic questions, current age, body mass and height, as well as the perceived changes in the physical activity questionnaire (PACQ) and the food frequency questionnaire (FFQ) covering the periods during the first and second lockdowns.

**Figure 2 biology-12-00326-f002:**
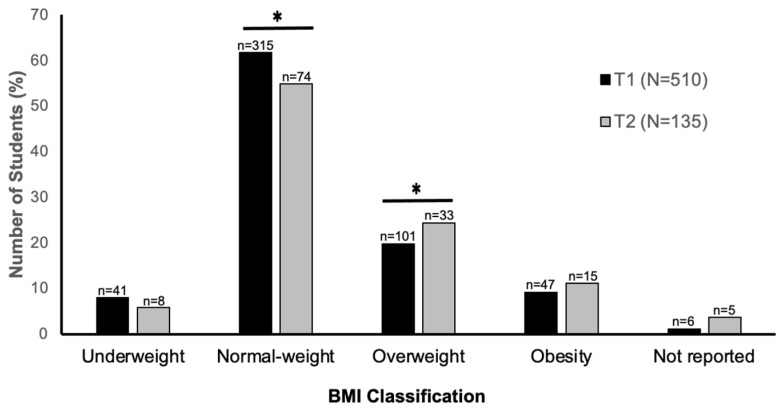
Percentage of Canadian university students meeting the criteria for underweight, normal weight, overweight, and obesity at the end of the first (T1) and second (T2) periods of the COVID-19 pandemic-induced lockdowns. * denotes significant differences in the percentage of students classified as normal weight and overweight between T1 and T2.

**Figure 3 biology-12-00326-f003:**
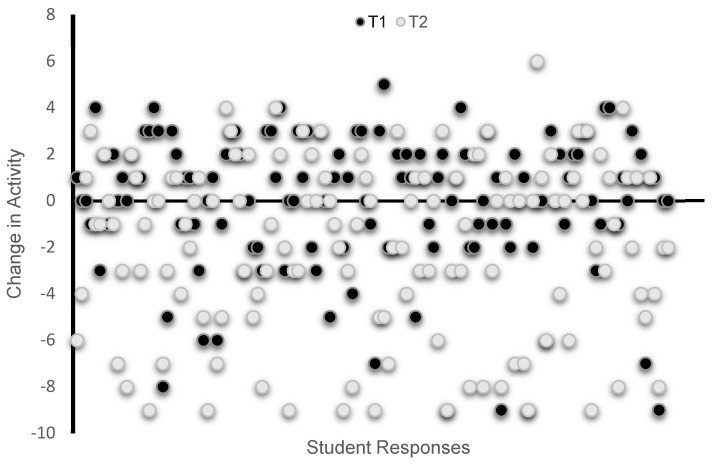
Perceived change in overall physical activity was self-assessed; at T1 and T2 participants were asked to indicate whether, relative to before the pandemic, their participation in each activity decreased (−1), increased (1), or did not change (0). Overall physical activity change scores were calculated based on the responses in all listed activities, including at-home workouts, walking, jogging, running, cycling, swimming, weight training, recreational sports, and sport-specific training.

**Figure 4 biology-12-00326-f004:**
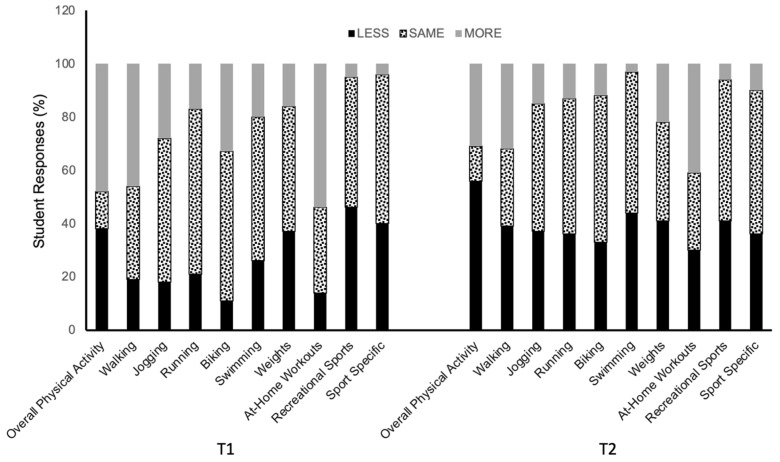
Mean percent frequency of perceived changes relative to before the COVID-19 pandemic in overall physical activity, and each of the specific activities during the first (T1) and second (T2) periods of lockdowns in Canadian university students (N = 135). Values are % frequency of responses.

**Figure 5 biology-12-00326-f005:**
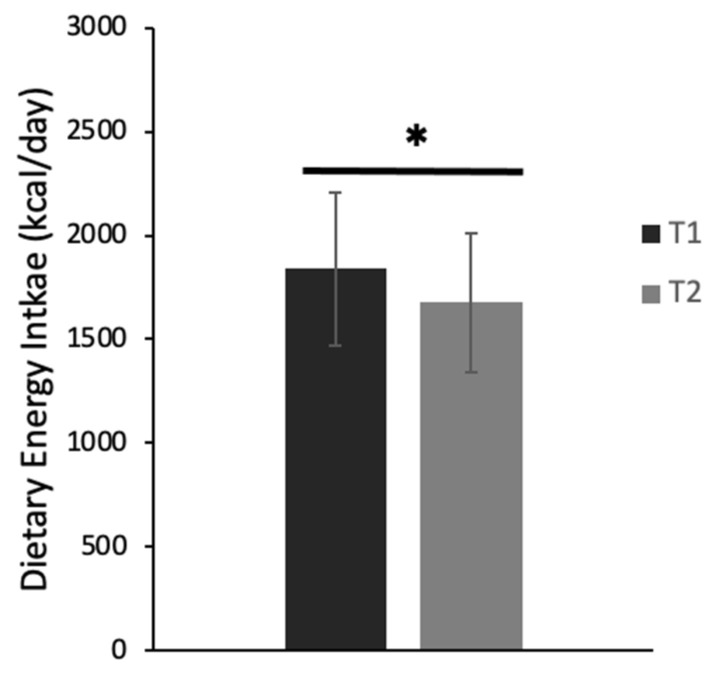
Daily energy intake during the first (T1) and second (T2) period of the COVID-19 pandemic-induced lockdowns/restrictions in Canadian university students (N = 135). Values are means ± SD. * denotes significant changes from T1 to T2 for the total cohort (*p* = 0.003, d = 0.27).

**Figure 6 biology-12-00326-f006:**
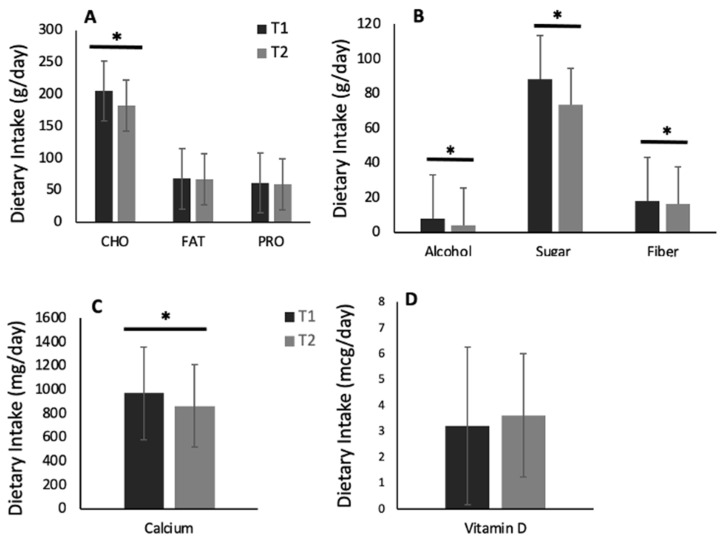
Macro and micronutrient intake during the first (T1) and second (T2) periods of the COVID-19 pandemic-induced lockdown/restrictions in Canadian university students (N = 135): (**A**) CHO = carbohydrates, FAT = fats, and PRO = protein; (**B**) alcohol, sugar, and fiber; (**C**) calcium; and (**D**) vitamin D. Values are means ± SD. * denotes significant changes from T1 to T2 (*p* < 0.05).

**Figure 7 biology-12-00326-f007:**
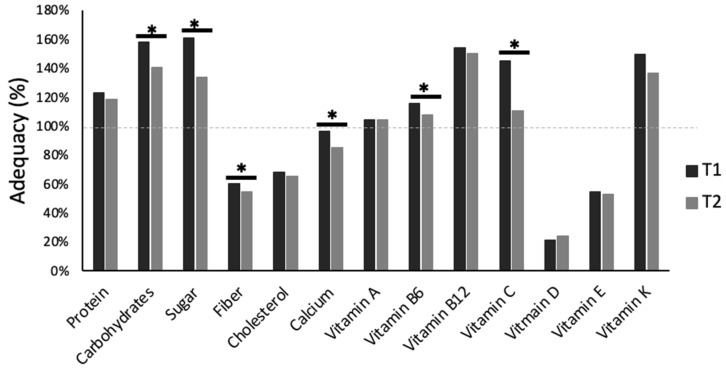
Macro- and micronutrient intake adequacy (% of recommended daily allowance) for the first (T1) and second (T2) period of COVID-19 pandemic-induced lockdown/restrictions for Canadian university students (N = 135). * denotes significant changes from T1 to T2 (*p* < 0.05).

**Table 1 biology-12-00326-t001:** Demographic characteristics of participants.

Demographics	T1 (September 2020)		T2 (March 2021)	
	N	%	N	%
Cohort (# of participants)	510	-	135	-
Age				
18–24	473	92.70%	121	89.60%
25–39	28	5.40%	9	6.60%
40+	9	1.70%	5	3.70%
Sex				
Male	99	19.50%	32	23.70%
Female	411	80.50%	103	76.30%
Race/Ethnicity				
White	375	73.50%	106	78.50%
Black	13	2.50%	4	2.90%
Arab	7	1.30%	0	-
Chinese	26	5%	5	3.70%
Filipino	12	2.30%	2	1.40%
Japanese	1	0.10%	0	-
Korean	4	0.70%	2	1.40%
Latin American	6	1.10%	4	2.90%
First Nations	2	0.30%	0	-
South Asian	34	6.60%	7	5.10%
Southeast Asian	8	1.50%	2	1.40%
West Asian	2	0.30%	0	-
Other	17	3.30%	2	1.40%
Prefer not to say	3	0.50%	1	0.70%
Year of Study				
Year 1	157	30.70%	23	17% *
Year 2	106	20.70%	26	19.20%
Year 3	126	24.70%	33	24.40%
Year 4	80	15.60%	23	17%
Year 5+	41	8.00%	19	14%
Graduated	0	-	11	8.1% *
Province/Region				
Ontario	402	78.80%	108	80.10%
West Coast (BC, AL)	71	13.90%	15	11.10%
Prairies (MB, SK)	9	1.70%	3	2.20%
East Coast (NB, NS)	17	3.40%	6	4.40%
Quebec	11	2.20%	3	2.20%
Living Arrangements				
*Living With*				
Alone	39	7.80%	7	5.20%
Family	390	76.30%	88	65.20%
Friends/Roommates	81	15.90%	40	29.6% *
*Living In*				
Apartment	71	13.90%	22	16.30%
House	348	68.30%	113	83.7% *
Other	91	17.80%	0	-

* denotes significant difference from T1; AL = Alberta; MN = Manitoba; SK = Saskatchewan; NB = New Brunswick; NL = Newfoundland; NS = Nova Scotia.

**Table 2 biology-12-00326-t002:** Body mass and BMI as self-assessed by the participants at end of the first (T1) and second (T2) pandemic lockdowns. Please note that participants with missing data were excluded from this analysis, resulting in a different number of included cases.

Variable		T1	T2	*p*-Value	Cohen’s d
	N	Mean (±SD)	Mean (±SD)		
Body Mass (kg)					
Males	30	84.0 (±20.8)	85.9 (±22.6)	0.056	0.18
Females	102	63.7 (±13.4)	69.2 (±18.5)	0.065	0.36
Total Cohort	132	68.3 (±17.5)	64.3 (±17.8)	0.008 *	0.23
BMI (kg/m^2^)					
Males	30	26.5 (±6.2)	27.1 (±6.6)	0.031 *	0.41
Females	100	23.2 (±4.5)	23.4 (±4.6)	0.065	0.16
Total Cohort	130	23.9 (±5.3)	24.2 (±5.1)	0.012 *	0.22

* denotes significant difference between T1 and T2; BMI = body mass index.

## Data Availability

Due to ethical restrictions, the data presented in this study are available to researchers eligible under the Research Ethics Board rules on request from the corresponding author.

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
