# Peer review of "Changes in Body Mass, Physical Activity, and Dietary Intake during the COVID-19 Pandemic Lockdowns in Canadian University Students"

_biology, 2023, doi:10.3390/biology12020326_

Round 1

Reviewer 1 Report

The Authors propose an interesting paper about dietary intake, physical activity and weight changes during the COVID-19 Lockdowns.

In the Methods section, the total student population of the universities contacted should be provided (eg 510 responders on a total population of how many students?).

In Table 1, a P value for T1 and T2 population should be provided (is there any significant difference in the sample of responders/lost to follow-up?)

The "acknowledgements" should be better defined.

Author Response

Comment: The Authors propose an interesting paper about dietary intake, physical activity and weight changes during the COVID-19 Lockdowns.

In the Methods section, the total student population of the universities contacted should be provided (eg 510 responders on a total population of how many students?).

Response: Thank you for giving us the opportunity to clarify. We have added the number of students who accessed the link and were considered potential participants versus how many completed the survey (lines 110-114). Although there were many universities contacted with a large population to include, we could only reach those who did read the announcement in specific forums. Thus, we do not know the percentage of the student population we reached out to.

Comment: In Table 1, a P value for T1 and T2 populations should be provided (is there any significant difference in the sample of responders/lost to follow-up?)

Response: Thank you for the suggestion. We added asterisks in the table denoting which percentages were significantly different between T1 and T2.

Comment: The "acknowledgements" should be better defined.

Response: We apologize for this oversight. We added an appropriate acknowledgments section (lines 481-483).

Reviewer 2 Report

This is a very well-written manuscript on an important topic.  The introduction nicely frames the problem and the gap in the literature.  The discussion is very good as well.  I just have a few suggestions. 

I am not sure if this can be done, but is there an estimate of the number of potential students that were reached online so a response rate could be approximated? 

Can you add the survey as a supplement or appendix?

How do the demographics compare to the population being studied?  No need for statistical analysis, but just a general view. 

Were the physical activity questions validated or developed by the investigators?

It would be interesting to know the number of those who gained and the number of those who decreased in BMI and the breakdown by sex/gender.  

It might be interesting to do a logistic regression to tease out risk factors for weight gain.   I think living alone and overall physical activity would be particularly interesting to see. 

Also, while the weight gain is modest overall, it might be very high in those who report decreased physical activity.  If possible, I would analyze the three physical activity groups separately. 

Do the authors have suggestions on how this issue should be approached in the event of another lockdown? 

Author Response

Comment: This is a very well-written manuscript on an important topic.  The introduction nicely frames the problem and the gap in the literature.  The discussion is very good as well.  I just have a few suggestions. 

Response: We appreciate the encouraging comments. Thank you.

Comment: I am not sure if this can be done, but is there an estimate of the number of potential students that were reached online so a response rate could be approximated? 

Response: Thank you for giving us the opportunity to clarify. We have added the number of students who accessed the link and were considered potential participants versus how many completed the survey (lines 110-114).  

Comment: Can you add the survey as a supplement or appendix?

Response: Thank you for the suggestion. The survey can be attached as suggested. However, we are not sure how so we just upload it as a separate file. In addition, not all questions of the survey were used in this specific manuscript, so we are not sure if this should be mentioned somewhere in the manuscript.

Comment: How do the demographics compare to the population being studied?  No need for statistical analysis, but just a general view. 

Response: We added comments on how representative our sample was or not, including age, province, sex, and ethnicity in the first paragraph of the discussion (lines 346-352) and in the limitations (lines 437-440). We also revised the conclusion in the abstract (line 40) and in the main text (line 453) to reflect our cohort.

Comment: Were the physical activity questions validated or developed by the investigators?

Response: The questions on the perceived change in physical activity were developed for the purpose of this study. Thus, these questions were not validated which was mentioned already as a limitation (lines 444-446). We also added an explanation in the methods as to why we used these questions (lines 173-180).

Comment: It would be interesting to know the number of those who gained and the number of those who decreased in BMI and the breakdown by sex/gender.  

Response: Thank you for this suggestion. The number of students meeting the criteria for each BMI class was reported in Table 1 with the demographics. However, we do realize that this is not the best way to report these numbers as they can be easily missed. Thus, we added a new Table 2, where we report all mean data for body mass and BMI both within each gender and in the total cohort. We then changed figure 1 to show the change in the number of participants classified under each BMI class. We hope this way is straightforward. We also run and added in the text the comparison analysis between males and females while controlling for the other variables; see statistical analysis (lines 219-221) and results (lines 237-241).

Comment: It might be interesting to do a logistic regression to tease out risk factors for weight gain. I think living alone and overall physical activity would be particularly interesting to see. 

Response: Regression analysis was run and added to the manuscript as suggested; see statistical analysis (lines 228-232), results (lines 314-337) and discussion (lines 386-390). 

Comment: Also, while the weight gain is modest overall, it might be very high in those who report decreased physical activity.  If possible, I would analyze the three physical activity groups separately. 

Response: Thank you for the suggestion. We did try the suggested analysis, but there were no significant differences between the three groups of physical activity change (less, more, no change).

Comment: Do the authors have suggestions on how this issue should be approached in the event of another lockdown? 

Response: We added some suggestions in the discussion, for example, students should aim to live with people and not be alone if a similar situation arises (lines 390-400). However, we would like to keep the discussion modest in offering socio-political suggestions beyond encouragement to be active and eat healthily.

Reviewer 3 Report

The present manuscript aimed to examine changes in body mass, physical activity levels and dietary intake during the COID-19 lockdowns in university students. While interesting, in my opinion the methodology to record data presents very important limitations. Therefore, results obtained should be very carefully interpreted.

Main comments

Participants

-The title report that participants in the study were “post-secondary students”. In line 114-115 it is stated that participants were undergraduate students. I suggest using the same term everywhere. Furthermore, this participants’ characteristic should be included in the abstract.

-Probably, listing the universities in the ethics section (lines 121-127 and repeated in lines 389-396) is not the best way to describe the origin of participants in the study.

-Number of participants, origin and distribution by sex. The total number of potential participants should be reported. Actually, taking into account the large number of universities participating in the study, the final number of participants seems very low. Even taking into account that only 135 participants completed the study, this leads to an average number of only 7-8 participants from each university. Is the whole number of participants representative of the total population? Which is the statistical power of the main results? Is the high percentage of women participants representative of the population?

Design

-In spite of only two measurement were performed, it would be good to include a flow chart of the study design, including the lockdown periods, and more specific questions. For example, were participants asked to report their body mass at the beginning or at the end of T1 and T2?

-In my opinion, the lack of a pre-lock down values should be included as a limitation of the study.

Data record and methods

-All data was not only self-reported but also, most of them was not current data at that time. Values supposed a recall from several months ago. Furthermore, “perceived changes” were asked. In my opinion, and in spite of it could be the only possibly, methodology used can not be considered as validated to perform this study, mainly taking into account the very low number of participants.

Discussion

-This section is mainly a repetition of results, with a brief comment on some of them. Furthermore, in my opinion, taking into account the methods used, most results are overinterpreted in the discussion. For example, those focused on the possible effect of micronutrient deficiencies determined using FFQ questionnaires from, at least, three months ago intakes.

Author Response

Comment: The present manuscript aimed to examine changes in body mass, physical activity levels and dietary intake during the COID-19 lockdowns in university students. While interesting, in my opinion the methodology to record data presents very important limitations. Therefore, results obtained should be very carefully interpreted.

Response: We appreciate the comment, and we improved the transparency of these limitations throughout the manuscript.

Main comments

Participants

Comment: The title report that participants in the study were “post-secondary students”. In line 114-115 it is stated that participants were undergraduate students. I suggest using the same term everywhere. Furthermore, this participants’ characteristic should be included in the abstract.

Response: Thank you for noticing. We changed it to university/undergraduate students throughout the manuscript as suggested. We also added the characteristics of the participants in the abstract (lines 26-27 and 29-31).

Comment: Probably, listing the universities in the ethics section (lines 121-127 and repeated in lines 389-396) is not the best way to describe the origin of participants in the study.

Response: We appreciate the comment, Table 1 includes what province/area the participants originated from.

Comment: Number of participants, origin and distribution by sex. The total number of potential participants should be reported. Actually, taking into account the large number of universities participating in the study, the final number of participants seems very low. Even taking into account that only 135 participants completed the study, this leads to an average number of only 7-8 participants from each university. Is the whole number of participants representative of the total population? Which is the statistical power of the main results? Is the high percentage of women participants representative of the population?

Response: Thank you for giving us the opportunity to clarify. The number of participants per sex and province is reported in Table 1. As suggested, we have added the number of students who accessed the link and were considered potential participants versus how many completed the survey (lines 110-114). Although there were many universities contacted with a large population to include, we could only reach those who did read the announcement in specific forums. Thus, we do not know the percentage of the student population we reached out to, but we agree that N=135 is low. Thus, for transparency, we added a comment on how representative our sample was or not, including age, province, sex, and ethnicity in the first paragraph of the discussion (lines 346-352) and in the limitations (lines 437-440). We also revised the conclusion in both the abstract (line 40) and the main text (line 453) to be specific to our cohort.

Design

Comment: In spite of only two measurements were performed, it would be good to include a flow chart of the study design, including the lockdown periods, and more specific questions. For example, were participants asked to report their body mass at the beginning or at the end of T1 and T2?

Response: Thank you for the suggestion, we added a flow chart with the timeline of data collection (figure 1). We also added specifics about each variable where relevant throughout the methods (lines 132-144; 159-161).

Comment: In my opinion, the lack of pre-lockdown values should be included as a limitation of the study.

Response: We agree. A statement has been added in the limitations (lines 447-448).

Data record and methods

Comment: All data was not only self-reported but also, most of them was not current data at that time. Values supposed a recall from several months ago. Furthermore, “perceived changes” were asked. In my opinion, and in spite of it could be the only possibly, methodology used cannot be considered as validated to perform this study, mainly taking into account the very low number of participants.

Response: We appreciate this criticism. To be transparent, we revised the limitations (lines 437-446) to clearly point out the small sample size, the retrospective nature of the survey, as well as the self-assessment and self-reporting of body mass and physical activity change.

Discussion

Comment: This section is mainly a repetition of results, with a brief comment on some of them. Furthermore, in my opinion, taking into account the methods used, most results are overinterpreted in the discussion. For example, those focused on the possible effect of micronutrient deficiencies determined using FFQ questionnaires from, at least, three months ago intakes.

Response: In order to slim the discussion and avoid overinterpreting our results, some parts were either deleted or revised, especially the paragraph regarding the dietary results (lines 401-436).

Round 2

Reviewer 3 Report

The manuscript has been lagely improved. However, taking into account the retrospective nature of the study, more references about the applicability of questionnaires and questions used shlould be included. Otherwise, in my opinion, validity of results would be to poor.

Author Response

Thank you for giving us the opportunity to justify our methods. We appreciate it. As suggested, we added references to the methods supporting the applicability of questionnaires and questions used. This applies to both demographics and physical activity sections. The food frequency questionnaire is a standardized questionnaire and references on each validity and reliability, specifically for the study population, were already provided.

Round 3

Reviewer 3 Report

The manuscript can be accepted in the present form.